# Carbon pricing and system reliability impacts on pathways to universal electricity access in Africa

Hamish Beath [1,2] ✉, Shivika Mittal [2,3], Sheridan Few [2,4], Benedict Winchester [2,5], Philip Sandwell [1,2], Christos N. Markides [5], Jenny Nelson [1,2] & Ajay Gambhir [2]

Off-grid photovoltaic systems have been proposed as a panacea for economies with poor electricity access, offering a lower-cost "leapfrog" over grid infrastructure used in higher-income economies. Previous research examining pathways to electricity access may understate the role of off-grid photovoltaics as it has not considered reliability and carbon pricing impacts. We perform high-resolution geospatial analysis on universal household electricity access in Sub-Saharan Africa that includes these aspects via least-cost pathways at different electricity demand levels. Under our "Tier 3" demand reference scenario, 24% of our study's 470 million people obtaining electricity access by 2030 do so via off-grid photovoltaics. Including a unit cost for unmet demand of 0.50 US dollars ($)/kWh, to penalise poor system reliability increases this share to 41%. Applying a carbon price (around $80/tonne $CO_2$-eq) increases it to 38%. Our results indicate considerable diversity in the level of policy intervention needed between countries and suggest several regions where lower levels of policy intervention may be effective.

Access to electricity remains a significant global challenge. Around 700 million people remain without access to any electricity supply, most of whom reside in Sub-Saharan Africa (SSA). Progress in reducing this number has slowed in recent years due to hardship from Covid-19 and the conflict in Ukraine[1,2]. Additionally, many with a connection experience a poor service level: globally, as many as 3.5 billion may live with unreliable power[3], causing a significant drag on economic growth[4]. Achieving universal household electricity access, a key component of the United Nations (UN) Sustainable Development Goal (SDG) 7[5], requires a focus on SSA and is an uphill challenge given the anticipated population growth and the number of people still without power in the region[2]. An urgent acceleration in progress is needed to retain any chance of achieving universal household access by 2030. Failing to do so will limit progress on several other SDGs[6], including improved healthcare (SDG3)[7], educational outcomes (SDG4)[8] and economic opportunities (SDGs 1, 8)[9].

The most appropriate technology to enable electricity access—grid extension, standalone systems, or mini-grids serving clusters of customers—varies with the context. Demand, population density, and distance to existing grid infrastructure are crucial factors influencing the cost competitiveness of different solutions. An argument for the use of solar photovoltaics (PV)-based off-grid systems can be made on cost, system reliability and environmental grounds[10,11]. Off-grid PV systems have become markedly cheaper, benefiting from the increased deployment of PV globally and the mass production of batteries for electric vehicles; capital costs for PV mini-grids fell by 65–85% in the decade up to 2019, and are expected to fall further by 2030[12]. Off-grid systems may offer consumers higher reliability and

[1]Department of Physics, Imperial College London, London SW7 2AZ, UK. [2]Grantham Institute—Climate Change and the Environment, Imperial College London, London SW7 2AZ, UK. [3]CICERO Center for International Climate Research, Oslo, Norway. [4]Sustainability Research Institute, School of Earth and Environment, University of Leeds, Leeds LS2 9JT, UK. [5]Clean Energy Processes (CEP) Laboratory, Department of Chemical Engineering, Imperial College London, London SW7 2AZ, UK. ✉e-mail: Hamish.Beath16@imperial.ac.uk

more predictable power than current national grid infrastructure in some contexts, which may encourage consumer preferences toward such systems[11,13]. Finally, PV-based off-grid systems can provide equivalent energy services with lower greenhouse gas (GHG) emissions relative to fossil-based alternatives[14]. Despite these advantages, in many low-access countries, renewables-based off-grid solutions are secondary in or omitted from electrification strategies[15].

Exploring different approaches for universal electricity access– e.g., centralised grid versus decentralised and fossil-based versus renewables-based energy infrastructure–has been done at a national and regional level using integrated assessment models (IAMs)[16,17] and at a more granular level using geospatial analysis[18–23]. Factors influencing which solution is cost-optimal (e.g., population density, distance from the grid) vary greatly at the sub-national level, and therefore, high-resolution geospatial analysis can provide useful guidance for planning cost-effective electricity access. Previous geospatial analyses have demonstrated the importance of electricity demand[19,23], climate-impacted cooling electricity demand[20], financing rates[22], and diesel pricing and subsidies[18] for least-cost pathways to electricity access. Geospatial analysis has been used to inform approaches to electricity supply beyond households, such as for healthcare facilities[24] and refugee camps[25].

Reliability of supply is an additional factor that should be taken into account when considering electrification approaches[26]. The reliability of supply is a key determinant of the *value* of access; with higher reliability enabling time-sensitive demands to be met[27]. Reliability is a key consideration impacting the costs of off-grid systems[28] and critical when comparing with national grids that often deliver poor service levels and associated negative impacts such as reduced security, reduced household income and additional expenditure on backup alternatives[29–31]. The reliability offered by different technology solutions has not been factored into previous geospatial work examining least-cost pathways to electricity access. This paper seeks to address this research gap through the inclusion of a cost for poor electricity service levels in the analysis.

Further, as countries and corporations seek to mitigate their GHG emissions and carbon pricing becomes more widespread[32,33], it is important to consider how it could impact pathways to electricity access. Voluntary carbon markets have the potential to provide significant amounts of climate finance for African countries, allowing them to sell carbon credits and spend money on clean energy access infrastructure[34]. Presently, Africa only produces a fraction of the carbon credits it could issue[35]. However, some existing schemes, such as the Distributed Renewable Energy Certificate (D-REC) initiative, already provide financial incentives for off-grid PV systems, increasing their uptake[36]. Given that a significant share of new connections required to meet universal access are unlikely to be financially viable without additional financial support, carbon financing may play an important role in expanding access[37]. Carbon pricing could influence the relative cost and shares of different access technologies and consequently be important for infrastructure planning. It has not yet been investigated using high-resolution analysis.

Here, we examine pathways to universal household electricity access in 43 SSA countries. Acknowledging the importance of considering country heterogeneity in the region's energy future[38], this paper combines detailed country-level data and electricity demand growth trajectories with open-source energy system modelling and high-resolution geospatial analysis to explore pathways to universal electricity access. In this work, we use a scenario-based approach and a modelling framework[39] that considers electricity demand growth by country (Fig. 1, see the "Methods" section); we examine both the reliability of supply and carbon pricing impacts on the shares of off-grid and grid provision of electricity at different demand levels. This paper addresses the following four main research questions regarding least-cost pathways to universal household electricity access in SSA.

Firstly, how and where do the shares of each technology change as the electricity demand level increases? Secondly, how does implementing a policy that includes a cost for units of unmet electricity demand change the share and spatial distribution of each technology used? Thirdly, how might implementing a carbon price change the shares and spatial distribution of the technologies used? And, finally, acknowledging the unevenness of existing policies and country heterogeneity, what is the sensitivity to different levels of policy intervention and how does this vary spatially? In addressing these questions, this paper aims to catalyse further research in this area and give a basis for implementing policies that can both expand access via low-carbon technologies and improve the *value* of access given to households from improved reliability of systems.

## Results
### Scenario design
To examine the least-cost pathways to universal household electricity access, we explore a range of scenarios that consider electricity demand, reliability of supply and carbon pricing (Table 1). We use population and electricity demand growth assumptions based on the shared socioeconomic pathway (SSP) 2[40] (see the "Methods" section). In our baseline assessment of the population requiring access, some countries are expected to reach universal household access (see the "Methods" section). Our least-cost scenarios include only the additional population above what we expect to occur in the baseline scenarios.

We use the Energy Sector Management Assistance Programme (ESMAP) Multi-Tier framework (MTF)[41] to guide demand levels, similar to other studies[19,22,42], and since many countries have electricity access targets aligning with the MTF. The MTF has Tiers 1–5, with household demand per year for each Tier of 4.5, 73, 365, 1241 and 2993 kWh, respectively. This paper examines only Tiers 1–4, with a focus on Tiers 2–4 for households that require electricity access. Tier 1 represents access only for phone charging or basic lighting and Tier 5 is largely unobtainable for newly connected households in SSA that have low incomes. In our scenarios, the MTF Tiers are used as the starting point for household annual demand. However, household demand trajectories vary depending on the expected gross domestic product (GDP) per capita growth by country (see the "Methods" section). Additionally, the estimated values represent electricity demand rather than demand met, which varies based on the reliability of different technology options. We do not use the MTF for blackouts or hours of service since we perform our own analysis on service reliability.

Our scenarios explore reference cases and reliability and carbon pricing variations (Table 1). Additionally, we perform spatially disaggregated sensitivity analysis on carbon prices and reliability penalty levels. Including sensitivity analyses, this work presents results from 110 scenarios.

### Progress toward universal household electricity access
In our baseline scenario, while some countries reach or make significant progress towards universal access by 2030, others see an increase in the number of people without access to electricity. Increases are driven by a combination of population growth and low levels of progress in specific regions (see the "Methods" section). The total number of people lacking electricity access in 2030 in the Baseline scenario for the 43 study countries is just over 470 million (from 505 million in 2020), with almost no further reduction by 2035 (Supplementary Table 1). This is in the context of a total population growth of 340 million between 2020 and 2030 and a further 240 million between 2030 and 2035. Higher-income countries such as Ghana, Kenya, South Africa, and Côte d'Ivoire are expected to reach universal access by 2030. By 2035, Senegal, Benin and Togo are also expected to reach universal household access. East African countries such as

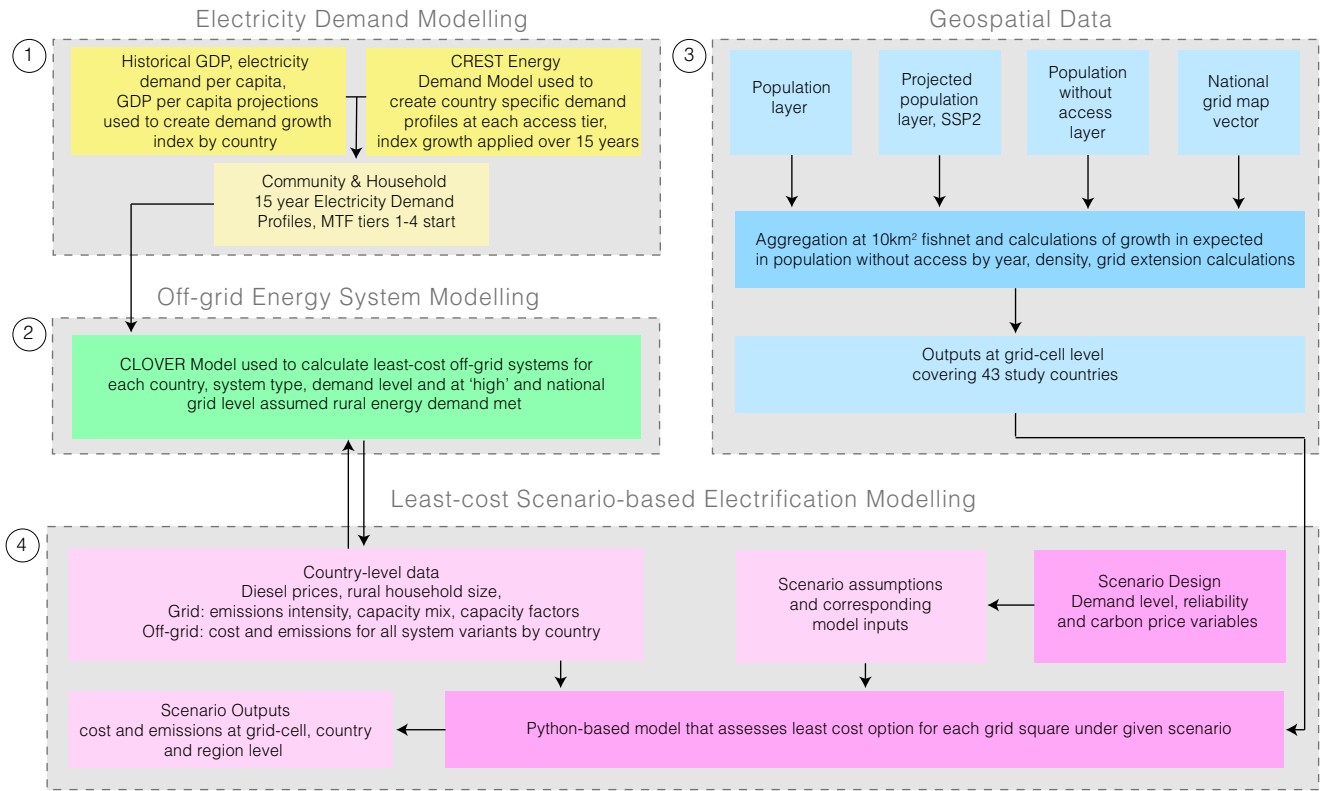

**Fig. 1 | Flow diagram explaining the methodological components for the analysis in this paper.** The first step was to model household and community level demand at different tiers of access with a growth rate applied by country linked to projected gross domestic product (GDP) per-capita growth. The second step was to model off-grid systems types at different demand levels for each country; this was done at different levels of energy demand met. The third part of the methods involved processing geospatial data covering the 43 study countries. The final step used the outputs from the first three steps in an open-source scenario-based least-cost electricity access model.

### Table 1 | The main scenarios modelled in this paper

| Scenario | Demand tier | Description | Target year |
|---|---|---|---|
| Baseline | N/A | Scenario that uses historical trends in improvements in access and projected population growth to estimate the population remaining without access by the target year. | 2020/2035 |
| **Reference** | | | |
| Ref_central | Tiers 1–4 | Central reference scenario where all unmet population above the baseline gain access via calculated least-cost electrification technology by 2030. | 2030 |
| Ref_single_modes | Tiers 3 & 4 | Reference scenarios where all unmet population above the baseline gain access via single access technologies, e.g., all via grid/diesel, etc., for emissions and investment comparisons. | 2030 |
| Ref_late | Tiers 2–4 | Central reference scenario where all unmet population above the baseline gain access via calculated least-cost access technology with a later target date. | 2035 |
| **Reliability** | | | |
| Rel_grid_all | Tiers 2–4 | Reliability scenario where all unmet population above the baseline gain access via calculated least-cost mode with the off-grid technologies sized at the assumed reliability of the rural grid in for each respective country. | 2030 |
| Rel_penalty_0.50 | Tiers 2–4 | Reliability scenario where all unmet population above the baseline gain access via calculated least-cost technology with a 0.5 US Dollar /kWh penalty for unmet demand applied across all (see the "Methods" section). | 2030 |
| **Carbon pricing** | | | |
| Ctax_median | Tiers 2–4 | Carbon price scenario where all unmet population above the baseline gaining access via calculated least-cost technology, with a carbon price scheme representing the median values from the Intergovernmental Governmental Panel of Climate Change Sixth Assessment Report (AR6) database[44] (C1 and C2 scenarios) is applied across all technologies (see the "Methods" section). | 2030 |

The baseline scenarios attempt to understand the expected outcomes given recent trends in connections and expected population growth. The Reference scenarios are variants of a least-cost pathway to universal household electricity access with no policy intervention. The reliability scenarios explore how the least-cost pathways may be different when sizing off-grid systems at lower levels or applying a financial penalty for unmet demand (see the "Methods" section). The carbon pricing scenario details the least-cost pathways to universal access when a dynamic carbon price is applied across all access technologies (see the "Methods" section).

Ethiopia and Uganda are expected to see considerable increases in population without access due to their high anticipated population growth that exceeds expected progress in connections (Supplementary Table 1, Supplementary Fig. 1).

### Electricity demand impacts least-cost technology option
Our results in the reference scenarios (Ref_central) imply that the electricity demand level is key in determining the most cost-effective technologies. Specifically, at the lowest demand level (Tier 1), off-grid

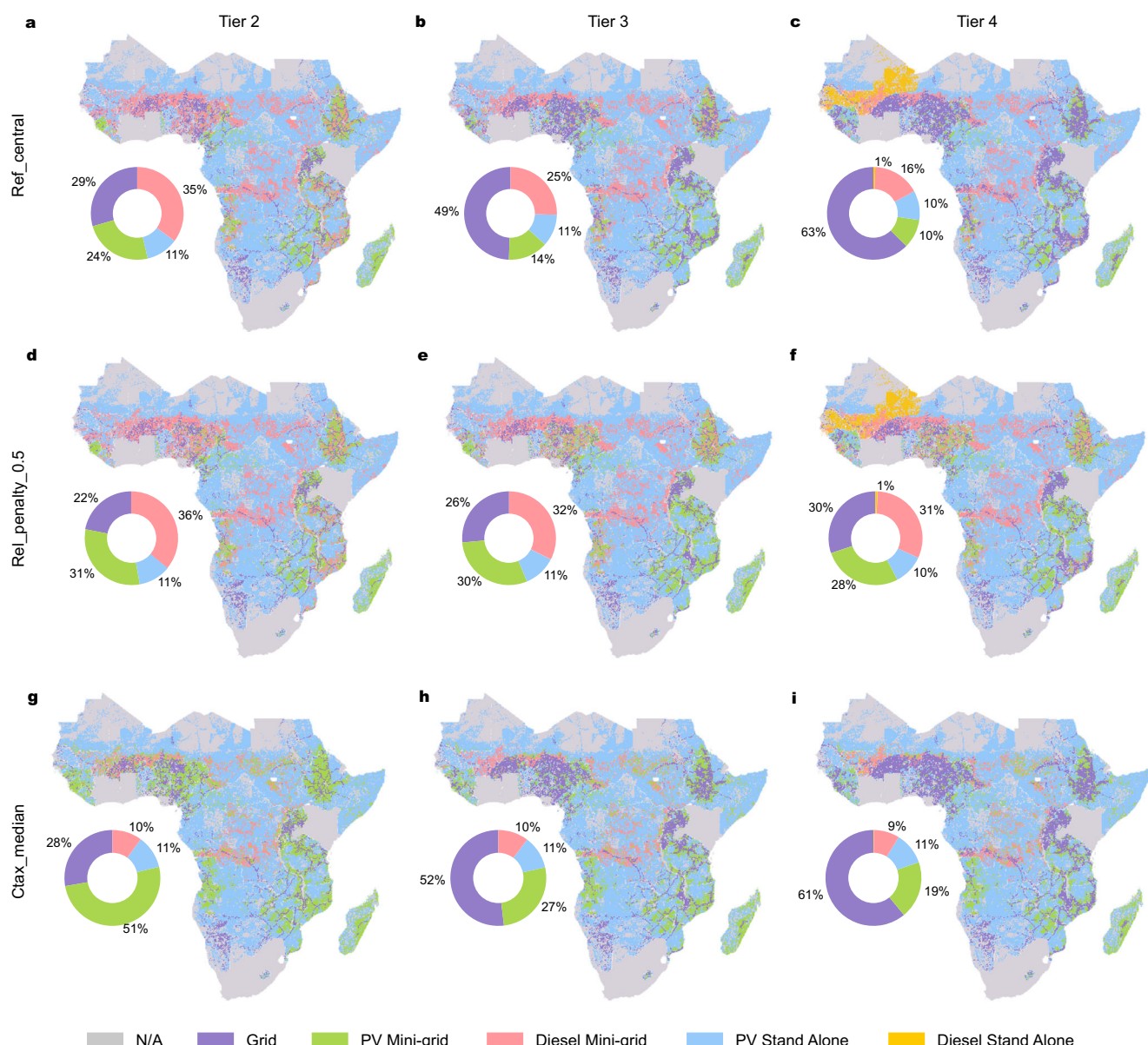

**Fig. 2 | Spatially distributed least-cost electrification technologies for the Ref_central, Rel_penalty_0.5 and Ctax_median scenarios at electricity demand Tiers 2–4.** Panels **a**–**c** represent the Ref_central results, with no carbon price or unmet demand penalty applied, they show a declining role for off-grid technologies as demand increases. Additionally, there are areas (e.g., parts of Central Africa) where diesel mini-grids are more prevalent; and other parts, such as countries on the eastern coast of Southern Africa, where PV mini-grids have greater usage. There are consistent areas of lower population density where stand-alone PV systems are almost always cheaper than diesel alternatives. Panels **d**–**f** represent the Rel_penalty_0.5 scenario applying a 0.5 US Dollars ($)/kWh of unmet demand penalty universally across all technologies, this reduces the share of the grid at all demand tiers, with the largest difference at Tier 4. Panels **g**–**i** represent the Ctax_-median scenario, equating to a carbon price scheme of around $4 in 2020, rising to $146 by 2030 here diesel usage is reduced in all three tiers, with the largest difference with Ref_central seen at Tier 2. US Dollars given in 2022 values.

PV-based technologies are the least-cost option for 78% of the population requiring access. At Tiers 2 and 3 (Fig. 2a and b), grid connections, as well as off-grid diesel, become cost-competitive. Diesel mini-grids are used in parts of Central Africa where the annual solar insolation is lower and there are few existing grid lines, as well as in parts of East and West Africa that have lower fuel prices (see Supplementary Table 6). At Tier 3 demand, the grid is the least-cost option for 49% of the population and at Tier 4 demand (Fig. 2c), the grid share increases to 63%. The share of off-grid PV installations falls from over a third at Tier 2 to 25% and 20% at Tiers 3 and 4, respectively. A delay in achieving the universal access target by 5 years (scenario Ref_late) increases the share of off-grid PV compared to a 2030 target due to lower assumed technology costs. For detailed results of this scenario (see Supplementary Note 2).

At lower demand Tiers, off-grid PV systems are relatively cheaper compared to other technologies as their modular nature permits sizing for low levels of demand. The cost of grid extension remains high regardless of the demand level, with only added grid capacity changing according to the demand Tier. Diesel systems are more economical above Tier 1 due to the minimum sizes and output capacity factors (see the "Methods" section).

**Including policy interventions for variation in reliability**
National grids have poor reliability in many countries, such as Nigeria and Zambia[43] (see Fig. 3b). Treating electricity access technologies equally regarding the levels of demand they meet may lead to poor outcomes for energy planning. We assess the impacts of the reliability

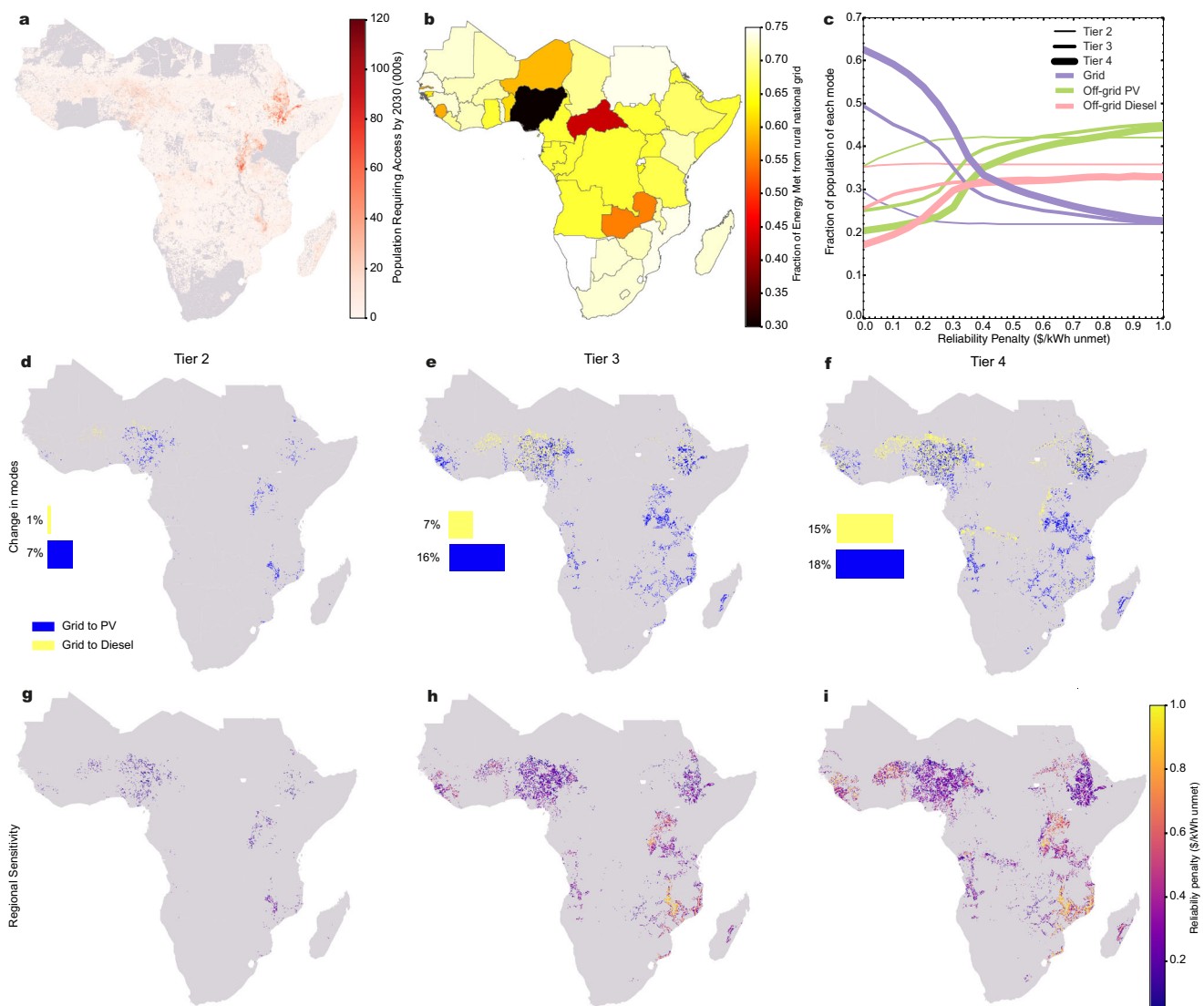

**Fig. 3 | Results for unmet energy penalty scenarios.** Panel **a** shows the spatial distribution of the estimated population requiring electricity access up to 2030 (see Baseline scenario). Panel **b** shows the estimated rural grid reliability by country used in the modelling, in percentage energy demand met (see Table S6 in Supplementary Materials). Panel **c** describes the sensitivity to different unmet energy penalties (0.05–1.00 US Dollars ($)/kWh unmet) that grid, off-grid PV and off-grid diesel shares have at each tier of electricity access. The plateaus that can be observed are due to the shares of the population without access that live in close enough proximity to the grid that the cost advantage at a given tier of access in terms of connection cost per household, even with a high unmet demand penalty, off-grid modes remain uncompetitive (Methods). Panels **d–f** describe shifts in the mode of access at Tiers 2–4 from the reference scenario (Ref_central) to the scenario applying a $0.50/kWh penalty. A greater impact can be seen at higher tiers, in large part due to the greater proportion of the population gaining access from the grid under the Ref_central scenario. Panels **g–i** show the different unmet energy demand penalty levels at which a change in electricity access technology occurs, Tiers 2–4. Increasing areas of shading can be seen as tiers increase due to increased shares of the grid used at higher tiers in the Ref_central scenario. US Dollars given in 2022 values.

of supply (% electricity demand met) on the technology shares and their spatial distribution. We first sized the off-grid systems in each country to match the assumed rural grid reliability in each country (Rel_grid_all scenario). Off-grid systems cost more at higher levels of electricity demand met[28], and so matching with unreliable grids sees substantially improved cost competitiveness. Detailed results of this scenario are provided in Supplementary Note 3.

Unreliable electricity supplies result in costs to households in SSA[30,31]. To represent this, we add a financial penalty ($/kWh unmet) across all technologies (Table 1, see the "Methods" section). A subsidy could be implemented instead, to have the same impact, see Supplementary Note 4 for further details. First, considering the scenario in which we implemented a 0.5 $/kWh penalty (Rel_penalty_0.5), the results (Figs. 2d–f and 3d–f) show that, at all Tiers, there is more usage

of off-grid technologies compared to in the Ref_central scenario, with the largest change being at Tier 4 demand. At a Tier-2 level of demand (Fig. 3d), the impact is small, with the majority of the change, 7% of the population, shifting away from the grid towards off-grid PV. This occurs in parts of East and West Africa where the grid is deployed at this Tier under the Ref_central scenario. A marginal shift from grid- to diesel-based systems is seen in parts of the Western Sahel, where diesel is the cost-optimal option at all access Tiers under the Ref_central scenario.

At a Tier 3 demand, bigger changes occur (16% more off-grid PV, 7% more off-grid diesel) than for Tier 2 demand when compared to the Ref_central scenario (Fig. 3e). Countries such as Nigeria and Senegal with low-grid reliability (Fig. 3b) see more use of off-grid PV. The results show a marginal impact for the reliability penalty in other

countries with poor grid reliability e.g., Zambia and the Central African Republic which already have higher levels of off-grid PV and lower levels of grid in the Ref_central scenario (Fig. 2b). Countries with higher grid reliability, e.g., Ethiopia and Tanzania, see differences in their pathways under Rel_penalty_50, with these countries having a higher share of the grid under the Ref_central scenario. At Tier-4 (Fig. 3f), a

large increase (18% of the population) in the usage of off-grid PV systems is seen in all parts of the continent deploying grid connections under the Ref_central scenario. Increased use of off-grid diesel under the Rel_penalty_50 scenario is the reason that we also see higher overall emissions, a 22% and a 4% increase for Tiers 3 and 4, respectively, when compared to the Ref_central scenario (Table 2).

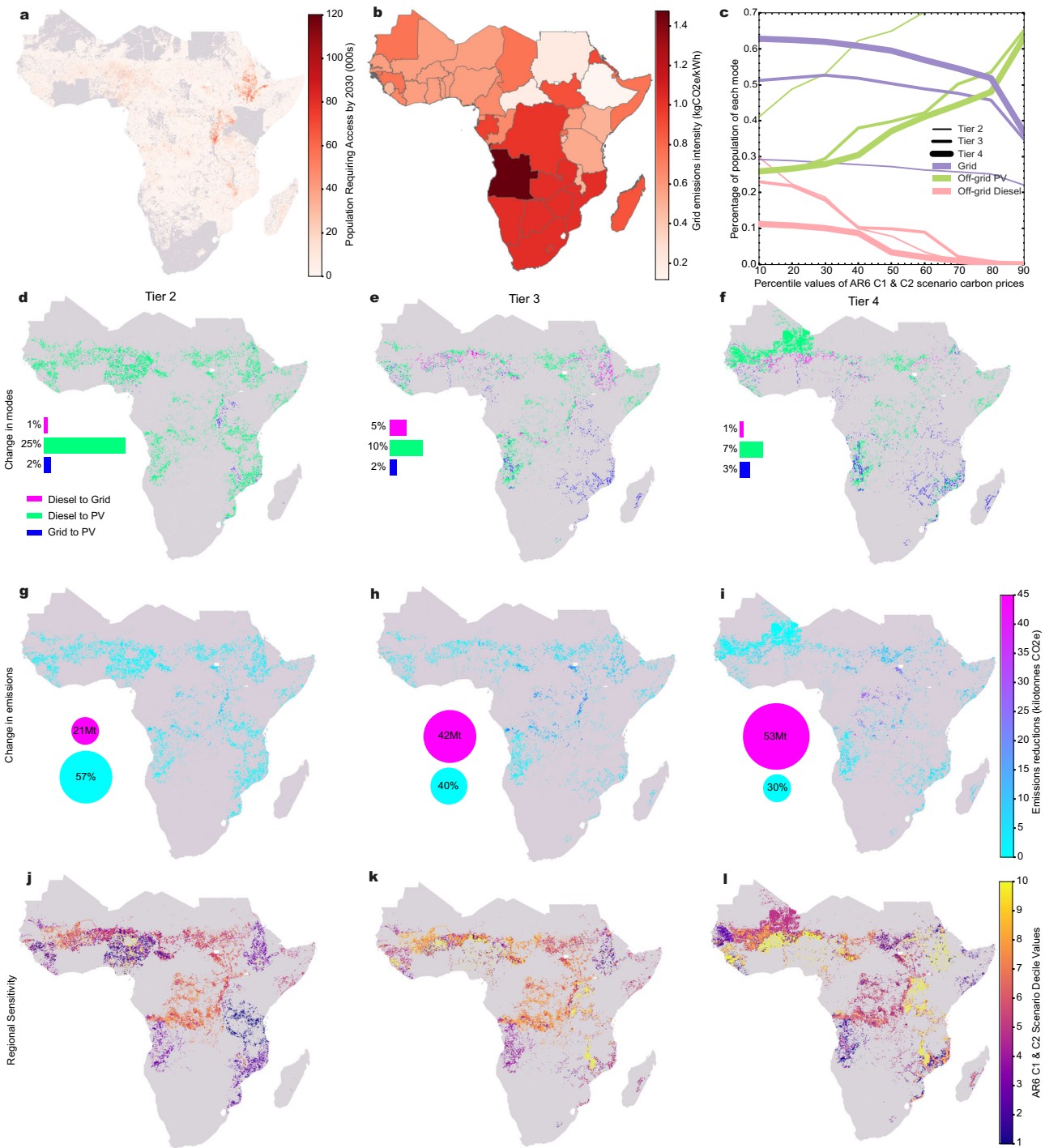

**Fig. 4 | Results for carbon price scenarios.** Panel **a** shows the spatial distribution of the estimated population requiring electricity access up to 2030 (see Baseline scenario). Panel **b** shows the grid emissions intensity of electricity (kilograms carbon dioxide equivalent per kWh) used in the modelling, detailed by country in Table S6 in the Supplementary Materials. Panel **c** describes the sensitivity to different carbon prices (see Table S5 in the Supplementary Materials) that shares of grid, PV and diesel have at each Tier of access. Panels **d–f** describe shifts in the electricity access technologies at Tiers 2–4 from the reference scenario (Ref_central) to the carbon price scenario (Ctax_median). Panels **g–i** describe the geographical distribution of the corresponding change in GHG emissions resulting from the carbon price scenarios at Tiers 2–4 (given in kilotonnes $CO_2$e, 1e7 on the scale). Panels **j–l** give mapped sensitivity showing at what carbon price (Intergovernmental Panel on Climate Change 6th Assessment Report database decile values) mode shifting occurs, Tiers 2–4.

We tested the sensitivity to different penalties between $0.05 and $1.00 at 5-cent increments (Fig. 3c). At Tier 2, the share of the grid drops from an already low level and plateaus after a threshold of 20 cents. Tiers 3 and 4 require higher penalties (20 and 50 cents) for meaningful changes. There is less impact after thresholds of 40 cents and 50 cents at Tiers 3 and 4, respectively. The level at which adding an unmet demand penalty influences the cost competitiveness between off-grid and grid technologies varies geographically at each Tier (Fig. 3g, h). For electricity demand Tiers 3 and 4, lower levels of penalty (below 0.4$/kWh) see differences in technology shares (when compared to Ref_central scenario) in certain countries, including Nigeria, Angola and Ethiopia. Changes do not occur until higher penalty levels (above 0.6$/kWh) in other areas, including parts of Mozambique, Rwanda and Uganda. This suggests that whilst national grid reliability contributes to geographical differences (Fig. 3b), other factors, e.g., the relative cost differentials between technologies in different regions, are important.

### Inclusion of a carbon price scheme

The implementation of a carbon pricing scheme (Ctax_median, equating to around $80/tonne $CO_2$-eq, see Table 1) sees notable changes in the technology shares and geographical distribution (Figs. 2g–i and 4d–f) when compared to the Ref_central scenario. The most significant change at all access Tiers is the reduction of off-grid diesel, predominantly supplanted by off-grid PV but also by the grid in some regions. The total share of the population changing least cost technology option in the Ctax_median scenario, compared to the Ref_central scenario, reduces as electricity demand Tiers increase.

Carbon pricing has the greatest impact on the uptake of off-grid PV systems at Tier 2. The share of the population that switches from off-grid diesel to off-grid PV systems is around 25% in the Ctax_median scenario compared to the Ref_central scenario. This occurs in all regions where diesel is deployed under the Ref_central scenario (Figs. 2a and 4d). A small change (2% of the population) is also observed from the grid to PV in parts of East Africa, including in areas of a relatively high density of population requiring connectivity (Fig. 4b). Movement from diesel to grid connectivity (1% of the population) occurs in West Africa where grid emissions intensity of electricity is moderate (around 0.6 $kgCO_2$-eq/kWh) and national grid lines already exist. The cost of grid extension with a carbon price becomes viable in these regions.

At Tier 3 electricity demand (Fig. 4e), under the Ctax_median scenario, there is less displacement of diesel by PV systems than at Tier 2 (10% of population) when compared to the Ref_central scenario. Shifts from the grid to PV systems occur at the same total level as Tier 2 (2% of the population); however, distributed over a more diverse area, particularly in countries with higher relative grid emissions intensity, e.g., Angola, Mozambique and Zimbabwe (Fig. 4b). There is a change (5% of the population) from diesel to the grid in parts of Ethiopia and Sudan that have a high density of population requiring access and lower grid emissions intensity. The results highlight that national grid emissions intensity influences the role of off-grid PV in achieving universal household electricity access when including a carbon price. For Tier 4 electricity demand, trends remain fairly similar to the Tier-3 results. However, there is both a smaller change from diesel to PV systems (7% of the population) and from diesel to the grid (1% of the population) in densely populated parts of East Africa (Fig. 4e).

Carbon pricing (Ctax_median) leads to reduced total emissions (2020–30) when compared to the Ref_central scenario: 37 to 16 $MtCO_2$-eq at Tier 2, 105 to 63 $MtCO_2$-eq at Tier 3, and 174 to 121 $MtCO_2$-eq at Tier 4. These numbers are small when compared to global totals. Reductions are concentrated in areas with a higher population density requiring access that sees changes in electricity access technology (Fig. 4a and g–h) and where the difference in emissions intensity between technologies is highest. This can be seen in central Africa, where there are pockets of high population density and changes from diesel to PV.

We explored sensitivity in carbon prices (Fig. 4c) from the IPCC Sixth Assessment Report (AR6) scenario database[44] (see the "Methods" section) between 10th and 90th percentiles at 10% increments (see Supplementary Table 5 for values). For Tier 2 and 3 electricity demand, carbon prices have more impact at lower levels than for Tier 4 demand, which sees virtually no change in total shares until the 40th percentile, after which there are reductions in both off-grid diesel and grid, with off-grid PV filling the gap. The results suggest that, generally, at higher levels of demand, a higher carbon price would be required to see changes. Looking at sensitivity to carbon pricing spatially (Fig. 4j–l), significant diversity can be seen regarding the effectiveness of carbon prices. Notably, at all Tiers of access, there are regions where a shift to different access technologies occurs at lower carbon prices (10th–30th percentile values). As the Tiers of access increase, the areas only sensitive to the highest carbon prices become more prevalent.

**Table 2 | The discounted investment costs (in 2022 US Dollars ($)) resulting in emissions and reliability metrics for selected scenarios**

| Scenario | Investment 2020-30 (billion $)[a] | Emissions ($MtCO_2$-eq) 2020–30 | Demand-weighted mean reliability |
|---|---|---|---|
| **Tier 3** | | | |
| Ref_central | 36 | 105 | 0.73 |
| Ref_single_mode (PV MG) | 51 | 14 | 0.90 |
| Ref_single_mode (Diesel MG) | 52 | 247 | 0.90 |
| Ref_single_mode (Grid) | 235 | 98 | 0.62 |
| Rel_Penalty_0.5 | 52 | 109 | 0.81 |
| Ctax_median | 42 | 63 | 0.73 |
| **Tier 4** | | | |
| Ref_central | 52 | 174 | 0.71 |
| Ref_single_mode (PV MG) | 78 | 24 | 0.90 |
| Ref_single_mode (Diesel MG) | 82 | 459 | 0.90 |
| Ref_single_mode (Grid) | 244 | 158 | 0.63 |
| Rel_Penalty_0.5 | 81 | 212 | 0.81 |
| Ctax_median | 62 | 121 | 0.71 |

The single mode scenarios represent meeting access with a single technology rather than a least-cost approach.

[a]Investment includes reliability penalties/carbon prices incurred in each scenario.

See methods for further detail on what is included and omitted from investment and emissions. $MtCO_2$-eq is megatonnes of carbon dioxide equivalent.

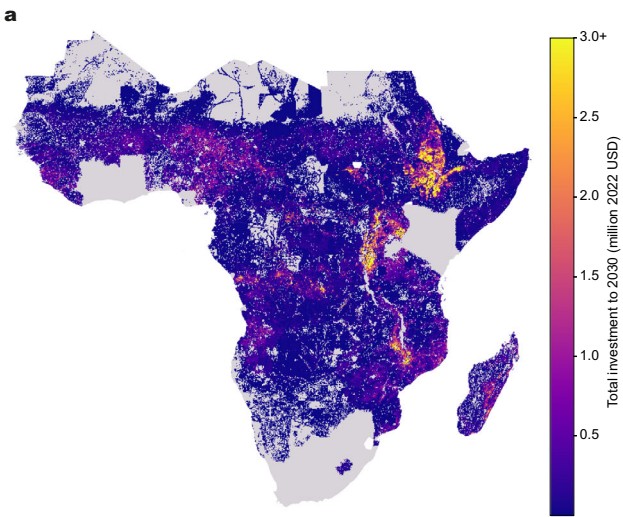
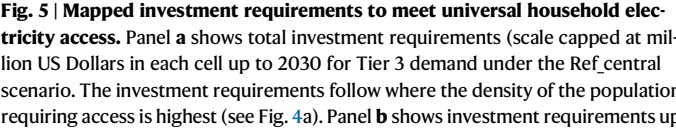
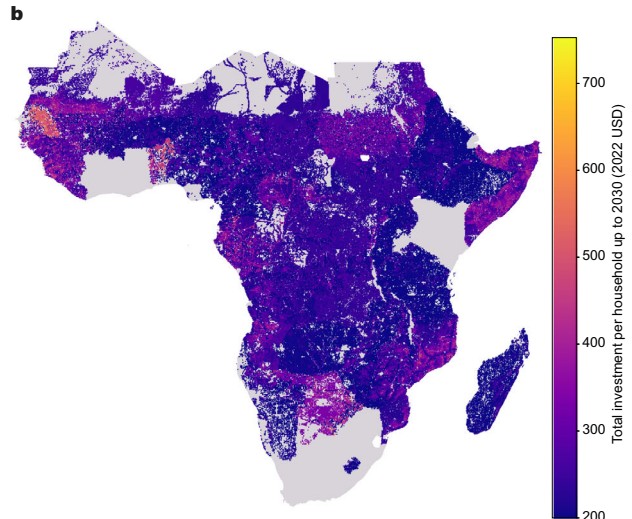

**Fig. 5 | Mapped investment requirements to meet universal household electricity access.** Panel **a** shows total investment requirements (scale capped at million US Dollars in each cell up to 2030 for Tier 3 demand under the Ref_central scenario. The investment requirements follow where the density of the population requiring access is highest (see Fig. 4a). Panel **b** shows investment requirements up to Tier 3 for the Ref_central scenario but per household. Here, the results are relatively uniform, with some variation based on country-specific factors such as grid coverage and composition, diesel fuel prices and electricity demand growth. US Dollars given in 2022 values.

## Increased investment for universal household access by 2030

Although our modelling does not capture all of the required investments needed for providing access (see the "Methods" section), it implies that taking a cost-optimal approach over one heavily dependent on certain technologies is advantageous (Table 2). Emissions contributions, even from the most polluting single-mode pathway are low when compared to global emissions and are occurring in regions with virtually no historical emissions contributions. However, there may be technology 'lock-in' from initial access technologies and emissions growth in the future as electricity demand grows. Therefore, the choice of electricity access mode may still be important for future emissions[45].

Geographically, the investment requirements broadly follow the population density requiring access at all Tiers (Figs. 4a and 5a). Under the Ref_central scenario (Tier-3 access) areas of the highest total investment requirement are in densely populated parts of central and eastern DRC, Ethiopia and northwestern parts of South Sudan. Other countries with significant areas of high investment needs are Malawi, Burundi and Uganda (Fig. 5a). Per-household investment requirements (Fig. 5b) at Tier 3 are more uniform, with higher values seen in areas with lower population density deploying standalone systems (Fig. 2b).

## Discussion

Population growth and the pace of progress in new household connections in SSA suggest there will be a failure to reach universal household access to electricity by 2030 or 2035. The strength of action and investment over the remaining decade will determine how close individual countries get to achieving SDG7. Following a least-cost pathway where a range of electricity access options (including PV minigrids and standalone systems) are built into electrification plans, can provide significant investment savings relative to plans that rely on incumbent technology options such as grid expansion or diesel generation. Following previous work[19], our results reiterate the importance of considering least-cost pathways in the context of electricity demand levels. For example, far greater emphasis is needed on off-grid PV at lower access Tiers. This point has salience for national electricity access plans that are centred around Tiers of access. Our modelling framework takes account of anticipated growth in electricity demand for households in each country; however, investing in infrastructure is likely to have impacts beyond the time frame considered here. Policymakers should consider possible technology lock-ins and possible associated costs for increasing access Tiers for households in the longer term. Our results highlight significant regional diversity regarding the least-cost technology deployed at different demand levels. As such, country-specific factors are crucial in assessing the role decentralised technologies will play in achieving the goal of universal household electricity access. High-resolution country-level studies can further elucidate the most appropriate pathways to inform national electricity access plans and the most appropriate associated policy support to achieve them.

Treating technology options with diverse levels of reliability as equal within electricity access plans is shortsighted. An urgent focus on the reliability of supply received by newly connected consumers is needed so they can move up the energy ladder and experience the economic benefits of electricity access. Accounting for the financial costs to households of poor service levels, extending a highly unreliable grid at a lower infrastructure cost than an off-grid solution with higher reliability may actually result in higher overall costs. Alternatively, sizing off-grid systems to meet the same service levels as rural national grids makes them more cost-competitive with national grid extension in some regions. When a cost for units of unmet demand is included in least-cost pathways, shares of access modes deployed change: at Tier 3, for example, applying a $0.50/kWh unmet penalty sees almost 80 million more people gaining access from off-grid PV. The impacts of such a policy intervention vary greatly depending on the electricity demand level and the level of penalty applied. There is also a high degree of spatial heterogeneity when observing whether there are changes in access technologies. A crucial observation is that there are areas with a high density of population needing access where applying a very low penalty for unmet demand sees uptake of off-grid technologies instead of grid connections. In practice, implementing a financial penalty for poor reliability, as is done in some high-income countries such as Norway, the Netherlands, and the UK[46,47], may not be feasible or desirable in countries where electricity supply companies are in a state of financial stress. Instead, offering an incentive scheme such as a performance-based subsidy for supply companies offering households a higher level of reliability may be a viable alternative. This has been demonstrated in Nigeria, where some mini-grid developers

are paid a subsidy per connection provided a minimum level of reliability is met[12,48]. Policymakers should seek to design mechanisms that can ensure reliable electricity access options are implemented and thus reduce the negative impacts felt by households from unreliable electricity grids in many SSA countries.

In this paper, we present a high-resolution geospatial analysis that explores carbon pricing impacts on pathways to universal household electricity access. Our modelling suggests that carbon prices may have a significant impact on the uptake of low-carbon off-grid technologies; however, it depends on the electricity demand level at the location being considered. At Tier 2, 140 million people switch to off-grid PV under the central carbon price scenario considered here, with a 50 million increase at Tier 3 demand levels. Whilst there is broadly a decline in the effectiveness of carbon price signals as electricity demand levels increase, there is significant variation in impacts spatially. Crucially, there are countries, e.g., Angola and Ethiopia, where relatively low carbon price signals lead to a movement to lower-emissions alternatives. This quantification of the spatial variation in the effectiveness of carbon price signals has implications for policymakers and stakeholders wishing to tailor policy interventions to specific countries or regions. As carbon markets become more prevalent globally, they may have a positive influence on the cost competitiveness of low-carbon electricity access solutions such as PV-based mini-grids[37]. Except for South Africa[49], countries in SSA presently do not have formal carbon taxation systems. However, other governments in SSA, such as Côte d'Ivoire, Botswana and Senegal are considering implementing schemes[33]. An additional consideration is the voluntary carbon market. There are already examples where low-carbon off-grid electricity access projects are being developed with the help of financing from carbon credits[34]. Schemes such as the Carbon Initiative for Development (Ci-Dev) and the D-REC initiative are facilitating carbon finance support for low-carbon electricity access projects[36,50]. Whilst not directly translatable from this analysis due to the complexities in carbon crediting schemes, our analysis points to the impact carbon credits may have, depending on the price level and where they are implemented. To maintain effective carbon markets and realise their possible benefits for energy access, countries in sub-Saharan Africa need to work to build regulatory, monitoring and verification systems[37]. If these can be implemented, our analysis highlights where schemes might be most effective for supporting lower-carbon electricity access.

The results in this paper emphasise the need for further regional and national-level studies. Addressing the lack of high-resolution spatial data regarding both the reliability of the grid and off-grid technologies would allow a more detailed comparison of supply options. In addition, expansion to non-residential demands would strengthen the analysis as electricity demand goes beyond households. Nevertheless, this study highlights the importance of considering reliability and carbon pricing when considering the role off-grid PV may play in developing pathways to SDG7.

## Methods
This study combines electricity demand modelling, energy system modelling, and geospatial analysis to investigate the impact of reliability of supply and carbon pricing on the shares of each technology used in scenarios aimed at achieving universal household electricity access (see Fig. 1).

### Electricity demand modelling
Household electricity demand levels are guided by the ESMAP MTF Tiers of Access[41]. However, we assume demand growth, underpinned by expected increases in GDP per capita growth and the historical relationship between GDP per capita growth and electricity per capita growth by country. For this, we use SSP2 'Middle of the Road' population and GDP growth projections. We reindexed long-range GDP per

capita growth projections from SSP2[51] using 2020 figures of recorded GDP per capita (PPP)[52], combined with IMF economic growth forecasts[53] that take into account the economic impact of COVID-19. From 2025 onwards, we applied the SSP2 growth rates to the adjusted data for the immediate future. We used historical GDP per capita (PPP) and electricity demand per capita data[52], available in varying quantities from 1970 to 2014, to derive a linear relationship between both variables by country. For countries without adequate data availability, a country with a similar socioeconomic profile in the same region was selected as a proxy (Supplementary Table 7).

We used the historical relationship calculated for each country as the income elasticity of electricity demand. We assume households do not begin to reach a satiation point for their electricity demand and, therefore, this value is held constant. Using the elasticity of the GDP per capita projections, we calculated an electricity demand growth index up to 2050, with 2020 as 1; to guide the increases used for the demand profiles. A range of other factors that may influence demand are not captured here; however, this approach rests on existing evidence suggesting income growth can lead to growth in electricity demand[54], including in low and middle-income countries (LMICs)[55].

To create relevant country-specific demand profiles to be employed in our off-grid modelling, we used the CREST open-source thermal-electrical demand model, which employs stochastic programming techniques to represent dwelling diversity and creates location-specific profiles that vary depending on temperature and daylight[56,57]. We generated 25 household electricity demand profiles for each country and for Tiers 1–4 of electricity access[42], as defined by the ESMAP MTF[41].

The household profiles were used to construct hourly, multi-year demand profiles that fit our annual future electricity demand estimates derived from the GDP per capita projections for each country, as explained above. For the mini-grid demand profiles, we assumed a community size of 100, and for each year, took a proportional mix of demand profiles from the electricity access Tier above and below the annual household consumption value, $A$, (in kWh) for a given year, $n$; with the annual household energy values (in kWh) represented by $T^U$ and $T^{Lo}$ for the upper and lower Tiers, respectively. We calculated the relevant percentages of the upper $T^U_{Sh}$ and lower Tier profiles $T^{Lo}_{Sh}$, with the upper given by:

$$T^U_{Sh} = \frac{100\left(A_n - T^{Lo}\right)}{T^U - T^{Lo}} \qquad (1)$$

and the lower Tier percentage $T^L_{Sh}$ the corresponding value that totals 100%. Using these shares, we added the respective proportion of household demand profiles for each Tier. We selected households (1–25) at random for each Tier profile added during this process. We used the one-year profiles for each country for relevant years to form 15-year demand profiles for each location that track the predicted demand growth with start years for every year between 2020 and 2035.

For modelling the standalone systems, we used a demand profile of 1% of the demand of the relevant mini-grid profiles. Our modelling considers energy demand on an hourly basis (see below) and, therefore, does not consider sub-hourly spikes in power demand. Consequently, the diversity benefits from the mini-grid demand profile passed to the single household profile are assumed to be minimal, with the daytime/nighttime shares of demand salient for system sizing.

### Off-grid energy system modelling
We used the CLOVER open-source energy system model[58] to estimate the cost and emissions of delivering electricity via different off-grid systems. The CLOVER model runs simulations of energy systems within specifications predetermined by the user. It has an hourly temporal resolution and is designed to simulate systems over a multi-year time horizon with dynamic demand, component health and

renewable resources. CLOVER interacts with the Renewables.ninja API to provide estimates of hourly solar generation for given coordinates, tilt and azimuth. Using these inputs, the Renewables.ninja model estimates solar output for a given system size based on reanalysis data from the MERRA-2 data set[59] (Supplementary Note 1).

The model has an 'exhaustive search' optimisation feature that tests numerous configurations to find the best-performing system for the optimisation criterion selected (see Eq. (2)). The model explores different system capacities in steps relevant to each technology type: for diesel, PV and battery technologies it explores capacities in units that these technologies are sold in. Diesel generators, for example, are not typically sold in units with capacities below 1 kW and so are less favourable to the smallest system types. Additionally, the model has a sufficiency criterion, typically a reliability level, and systems that do not meet this are not considered. The model can break down the system life into multiple periods to reflect the need to add capacity when demand increases and components degrade.

We used the model to find the cost-optimal standalone systems for single households and mini-grid systems powering 100 households, at each of our locations using solar PV and battery systems, as well as diesel-powered systems. We assumed a 15-year project length, but re-optimising systems every five years to take into account the growing demand profiles and degrading components. For our reference scenarios, we set our sufficiency criterion as 90% of energy demand met. For simplicity, we assume consistency in the type of components used across system sizes, with them scaled as necessary by the model for the demand level (technical inputs used are given in Supplementary Table 2).

We optimised systems for cost, given this is most representative of how systems are designed in reality. We use the levelised cost of *used* electricity (LCUE) as our metric of unit cost. This metric is similar to the more commonly used levelised cost of electricity (LCOE); however, it considers only electricity used rather than generated and is, therefore, more appropriate for off-grid systems that have significant dumped energy[60]. The LCUE is calculated by dividing the discounted sum of the total costs (capital investment $I$, operation and maintenance $M$, and fuel $F$) in each year, by the electrical energy used $E^U$ (also discounted) in each year:

$$\text{LCUE} = \frac{\sum_{n=1}^{N} \frac{I_n + M_n + F_n}{(1+r)^n}}{\sum_{n=1}^{N} \frac{E_n^U}{(1+r)^n}} \qquad (2)$$

For a full breakdown of the cost and emissions inputs used (see Supplementary Table 3). Associated total and unit emissions for the system are given as outputs for each system.

For each of our countries, we used the model to find the cost-optimal mini-grid and standalone systems with starting electricity demand Tiers (1–4), 15-year system lifetimes and start years of 2020, 2025, 2030 and 2035 to reflect growing demand and reduced component costs. Further, systems were modelled at 'high' reliability (90%) of electricity demand met, and 'grid' reliability; which equates to the estimated rural grid reliability in each country (Supplementary Table 6). The emissions and cost outputs of this process are used in the following stage.

## Geospatial data

For our analysis, we used geospatial data sets covering Sub-Saharan Africa (43 countries) for population, projected population growth, population without access to electricity, national grid infrastructure and national boundaries. Population gridded datasets for the years 2014 and 2020 were taken from the LandScan Global data sets[61,62]. Downscaled population projections (SSP2) at 10-year intervals were used (2020, 2030 and 2040) for spatially disaggregated population trajectories[63,64]. We used a gridded data set providing estimates for the population without access to electricity across Sub-Saharan Africa based on remote sensing night-light data from Falchetta (2019)[65]. We used the data from both the years 2014 and 2020[66]. We used vector maps of the African grid available from Moner-Girona and Georgia (2021)[67]. We used an open-source data spatial layer for national boundaries from IGAD Climate Prediction and Applications Centre (ICPAC)[68]. We aggregated these data and performed relevant calculations for our own gridded map at a 10 km² resolution to be used in our modelling. We use the term 'grid cell(s)' to refer to the individual parts of the mapped area we analyse for this research.

## Estimation of baseline population without access

To calculate baseline estimates of the population without access for the years 2020–35 we first used the historical geospatial electricity access population data (2014 and 2020) from Falchetta (2021)[66] to produce estimated annual changes in the population without access to electricity at the grid cell level. We also consider expected population growth at the grid cell level. For grid cells with partial electricity access, we assume in our forward projections that population growth in each cell is split between people with and without access to electricity based on the shares from the 2020 data in each cell. On this basis, we calculated annual estimations of the population without access at the grid-cell level up to 2035. This calculation considers assumed population growth amongst the population without access and any expected change in access, as per the historical trend. For a grid cell, access improvements may mean it reaches universal access prior to the end of the modelling time horizon. If this occurs, this annual progress is considered elsewhere (within country borders). We aggregate it by year at the country level and this amount is then used and allocated to provide access to households within grid-cells without access, starting with grid-cells with the highest number of people without access first. This process is recorded by year for use in the next section. This logic rests on the assumption that an investor or government agency would seek to provide access to the areas with the highest population density required first. We assume that these improvements in electricity access stay within national borders and once a country reaches universal access, it is removed from the modelling process.

## Estimating population trajectories for universal access

To reach universal household access in all countries not expected to achieve universal access by the target year under the baseline scenario, a pathway must be defined. This is done on a country-by-country basis, with the annual number of people connected for each country, $P_X^n$, defined by

$$P_X^n = P_R^n \left( \frac{n}{N} \right)^2 \qquad (3)$$

where $P_R^n$ is the remaining population for a given country in year $n$, and $N$ is the total number of years. This produces an S-curve, leading to most connections made in the middle of the period[69]. The model then takes the approach of connecting cells by the year in each country up to the amount $P_X^n$. Any growth in the non-electrified population in already 'electrified' cells is also taken into account first, after which the next densest non-connected cells are added first. This rests on the assumption that an investor or government agency would seek to provide access to the areas with the highest population density required first. Cells that are registered as being connected under the baseline scenario are skipped, avoiding double counting. The trajectory of the population reaching universal access (by individual map cell) is then converted to the number of households using the rural household size by the country for the most recent year available from the Global Data Lab average household size database[70] (Supplementary Table 6).

## Calculating least-cost technology pathways

We devised a modelling framework to assess the least-cost technology options under different scenarios[39]. A detailed mathematical breakdown is given in the Supplementary Methods.

For national grid costs, the distance from the nearest grid line to the centroid of the selected cell, combined with an estimated cost per km of grid extension and a grid connection fee per household is applied for non-generating infrastructure costs. The installation of equipment such as transformers or substations is omitted. For additional capacity, we derived values for weighted average grid capacity factor $G_y$ (%) and cost in 2022 USD $G_I$ for units of capacity (kW/kWp) on the grid in each country $\kappa$ using generation mixes per country[71,72], and country-relevant capacity factors and costs[73] (Supplementary Table 6). With these values, the model estimates additional grid capacity, $G^{cap}$, required in each cell, ($\phi$, based on the average hourly electricity demand $E^L$ (in kWh) in the final year $N$; taking into account the assumed reliability of the rural grid (estimated % of energy demand met) in each country $G^R(\kappa)$ and is given by

$$G^{cap}(\phi) = \left(\frac{1}{G_y(\kappa)}\right) E^L_N \left(G^R(\kappa)\right) \qquad (4)$$

The estimated additional grid capacity for each grid cell $G^{cap}(\phi)$ is then multiplied by the estimated cost per installed unit of capacity for the relevant country $C^I_G(\kappa)$. The generating and network costs are discounted according to the year of construction in the model, as guided by the above (see Supplementary Methods for further details).

Data on grid reliability in SSA is sparse. However, estimates of varying recentness for grid reliability by country are available from the World Bank Enterprise Survey[43] for business connections, typically in urban areas, which are known to be considerably higher than those in rural locations[14,74]. Using the average number of outages and average length, we estimate the percentage of energy unmet annually (assuming a flat demand curve). We then assume a 25% further penalty to reflect that supply on national grids is poorer for rural connections and worse for households than nearby enterprises[75].

To calculate grid emissions, we use the demand estimations (described above) by country and by Tier. Grid emissions intensity figures are assumed to be constant (Supplementary Table 6). The reliability factor, as above, is applied to the demand totals before calculating emissions resulting by multiplying by the relevant grid emissions factor. We do not include embedded emissions in the generating capacity due to poor data availability. We include embedded emissions in grid extension infrastructure, with the emissions split over the assumed infrastructure lifetime (Supplementary Table 4 and Supplementary Methods).

For the off-grid systems, emissions and investment figures have been calculated for system type and country using the CLOVER model (see Off-grid energy system modelling). The outputs are used to calculate estimates of the cost and emissions of electrification via each mode, for each cell; to be used in the final stage. The number of households requiring connection is multiplied by the investment estimate (per household) for each given system: diesel and PV mini-grid and standalone systems, sized for each country, start Tier of access; and levels of reliability. These numbers are then discounted (at 8%) according to the respective year of connection. Emissions for off-grid PV are split annually over the assumed system lifetime (Supplementary Table 4 and Supplementary Methods).

In the final stage, estimations of the least investment cost approach are made for each cell covering all countries requiring electrification to reach universal access by 2030. Cells are first assessed for population density to establish whether standalone or mini-grid systems are more appropriate (see Supplementary Methods). Next, the grid and the relevant off-grid PV and diesel systems investment requirements are each compared by cell covering the entire gridded

map. Using the estimated investment required for access via each mode, the least cost approach for each is selected for the given scenario. For scenarios applying a carbon price or reliability penalty, these are applied uniformly across all modes to tonnes of $CO_2e$ and kWh of lost load, respectively, and discounted prior to finding the least-cost approach.

For scenarios accounting for the financial cost of unmet demand, a penalty is added across all technologies for each lost kWh, acknowledging that there is an economic cost for households in SSA when power outages occur[30,31]. Often referred to as a 'loss of load penalty' existing studies have focused on high-income countries, finding large variation but with higher costs for businesses than households and the methodology of calculating the value having a large impact[76]. For our study, this value is difficult to quantify consistently and will vary across countries, income groups and demand levels. To address this uncertainty, we examine sensitivity to different penalty levels regionally and at different electricity demand Tiers of access. The values we investigate (5 cents–$1) are considerably lower than those suggested for high-income countries[76]; however, they are in line with the cost of a unit of electricity, considering only households, and our study is focused on countries and households where incomes are typically very low. We use 50 cents as our main scenario as it is the threshold at which significant change in technologies is seen at Tiers 2–4. The penalty amounts are included in the total investment costs, and would hypothetically be incurred by an electricity supply company when blackouts occur.

The carbon prices used are values for 2020, 2025 and 2030 from the IPCC AR6 C1 and C2 scenarios which correspond to a 50% chance of limiting warming to within 1.5 degrees by 2100 without, and with overshoot respectively. We use values converted into 2022 USD provided from the AR6 scenarios database[44], the values by percentile are given in Supplementary Table 5.

## Limitations

Our methodology has several limitations that are outlined here, with the anticipated influence on our results. Firstly, for the grid emissions intensity, reliability and costs by country are assumed to be constant. In reality, they are likely to change over time. Broadly, we can expect grids to become less emissions-intensive over time as countries add more renewable-based generating capacity. However, this is less certain for lower-income countries in SSA. In addition, there is uncertainty around grid reliability and the cost of additional capacity over time. In some countries, grid reliability is likely to improve as the grid is strengthened; however, others may continue to see low reliability. Improvements or deterioration in values for the grid in each country are likely to make the grid more or less favourable depending on the variable and scenario.

In addition, our modelling framework has a simplistic consideration of grid extension. We use the distance from the grid combined with a single extension cost per km for each cell as a proxy for the cost of extending. We do not take into account substations, transformers or other grid infrastructure needed. This is a limitation to our work as we are likely misinterpreting the costs of the grid in certain regions. A grid-routing algorithm, as is used in other work[21], would likely lower the cost of grid extension for some cells, whereas including other grid extension equipment would increase it. Using a grid-routing algorithm is computationally intensive and would prohibit an annual temporal resolution with many scenarios and sensitivities as is required for this study.

Finally, our emissions accounting method does not capture the embedded emissions in the added national grid-generating infrastructure. This is largely because it is highly complex to do so and the lack of data availability for plants installed in the region. This omission will give an emissions reduction for grid-generated electricity and, therefore, create a bias towards the grid in carbon pricing scenarios.

**Reporting summary**

Further information on research design is available in the Nature Portfolio Reporting Summary linked to this article.

## Data availability

All data used for running the scenario analysis and for producing figures can be found at the Github repository along with the code for the geospatial tool, at github.com/hamishbeath/LEAF-geospatial-energy-africa. The data (with the code) is archived with Zenodo https://doi.org/10.5281/zenodo.10793959.

## Code availability

The model[39] used to produce the scenario outputs are available at github.com/hamishbeath/LEAF-geospatial-energy-africa; the CLOVER model code is available at github.com/CLOVER-energy/CLOVER and has documentation available at github.com/CLOVER-energy/CLOVER/wiki.

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

## Acknowledgements

The authors would like to acknowledge the following funding sources: UK Engineering and Physical Sciences Research Council (EPSRC) RENGA (EP/R030235/1); SUNRISE (EP/P032591/1), ATIP (EP/T028513/1) and Research England GCRF QR Funding, UK (H.B., S.F., P.S. and J.N.); EPSRC, UK (EP/X52556X/1) (H.B., B.W. and J.N.); Natural Environment Research Council (NERC), UK (NE/R011613/1) (H.B. and J.N.); NERC, UK

(NE/2451429) (B.W. and C.M.); UK Research & Innovation for funding, under Project No. 1004545, as part of the Horizon Europe 'IAM COMPACT' Research and Innovation Project (10105630) (S.M. and A.G.); the Royal Society for the award of a Research Professorship (J.N.). The project funders were not directly involved in the writing of this article. The authors would like to thank Dr. Iain Staffell for comments given on an earlier draft of this work.

## Author contributions

H.B., S.M., S.F., P.S., J.N. and A.G. conceived and designed the study. H.B. performed the experiments and analysed the data. H.B., S.M., S.F., B.W., P.S. and C.M. contributed materials or analysis tools for the study. H.B. led the writing of the study, with writing contributions from S.M., S.F., B.W., P.S., C.M., J.N., A.G., H.B. and S.M. S.F., B.W., P.S., C.M., J.N. and A.G. reviewed the manuscript before submission.

## Competing interests

The authors declare no competing interests.
