## [Peer review file · Nature Communications]

REVIEWER COMMENTS

Reviewer #1 (Remarks to the Author):

The manuscript focuses on an important aspect of providing high level of energy access in SSA: how to change the low pace of increasing access, and what are the optimal technology mixes in providing access. As the progress in increasing electrification to the population has slowed down postCovid period, the policy options to go back to the positive track becomes important. The authors use the available GIS based analytical tools in a robust way, and communicate the results and the sensitivities convincingly.

The authors have analysed the impacts of 2 policy options on the electrification pathways: 1) applying penalties for unmet demand and 2) applying carbon pricing.

The analysis shows both have strong impacts both on the electrification rates and strong positive impact on the optimal technology mix from climate emission point of view.

The key question is what is reality in the policy agenda of applying the above mentioned policy instruments in the given timeline in the SSA development setting. This is not only a critical question as the application of these policy instruments are not uniform even in the developed economies setting (there are some well functioning Carbon taxing ie. Canada, Japan some European countries and Carbon trading schemes ie. in EU), but some can argue that adding a penalty to the under-financed state owned energy companies would simply amplify the barriers in the electrification process. If the authors could provide some examples in some SSA countries (or from other developing countries) where these instruments were tried or introduced, it could increase the research added value to the policy agenda and decision making as well as to the ongoing research discussion. Otherwise the well prepared analysis has the risk of remaining only an academic exercise.

Providing these policy use cases I would strongly recommend the article to be published.

Reviewer #2 (Remarks to the Author):

The manuscript titled 'How reliability and carbon prices impact pathways to universal electricity access in Africa' addresses an important and relevant topic. However, there are several areas where the manuscript needs improvement to enhance its clarity and overall quality.

The title of the manuscript, "How reliability and carbon prices impact pathways to universal electricity access in Africa," does indeed sound like a question. It would be more appropriate to rephrase it into a statement or summary of the study's objective. For example, "Assessing the Impact of Reliability and Carbon Prices on Achieving Universal Electricity Access in Africa."

One of the primary issues with this manuscript is its lack of coherence and proper organization. The structure of the paper needs to be improved for better readability and comprehension. It is recommended that the authors consider restructuring the paper as follows:

Introduction: Provide a clear introduction that outlines the importance of the study, its objectives, and the research question.

Literature Review: Expand the literature review section to provide a more comprehensive overview of existing research in this area. This will help readers understand the context of the study.

Methodology: The methods section should be expanded to provide sufficient detail for the work to be reproduced. Explain the data sources, variables, and statistical methods used in the analysis.

Results: Present the results in a logical sequence and use clear headings and subheadings to guide the reader through the findings. Visual aids such as tables and figures should be used to enhance understanding.

Discussion: Analyze the results in the context of the research objectives. Discuss the implications of the findings and how they relate to existing literature.

Conclusion: The manuscript lacks a conclusion section, which is essential to summarize the key

findings and their significance. Provide recommendations for future research or policy implications. References: Ensure that all references are cited correctly and consistently throughout the manuscript. In conclusion, while the topic of the manuscript is noteworthy and relevant, it requires significant revisions to improve its overall quality. The authors should focus on enhancing the structure and coherence of the paper, providing sufficient detail in the methods section, and rephrasing the title to make it more declarative. These revisions will enhance the manuscript's readability and impact. I recommend that the authors carefully address the above-mentioned issues and consider incorporating the suggestions into their revised manuscript.

Response to Reviewers: NCOMMS-23-28925-T

#	Comment	Response
1	Reviewer 1 The manuscript focuses on an important aspect of providing high level of energy access in SSA: how to change the low pace of increasing access, and what are the optimal technology mixes in providing access. As the progress in increasing electrification to the population has slowed down post Covid period, the policy options to go back to the positive track becomes important. The authors use the available GIS based analytical tools in a robust way and communicate the results and the sensitivities convincingly.	We thank the reviewer for their feedback and are encouraged that they feel the results and sensitivities are communicated clearly.
2	The authors have analysed the impacts of 2 policy options on the electrification pathways: 1) applying penalties for unmet demand and 2) applying carbon pricing. The analysis shows both have strong impacts both on the electrification rates and strong positive impact on the optimal technology mix from climate emission point of view. The key question is what is reality in the policy agenda of applying the above mentioned policy instruments in the given timeline in the SSA development setting. This is not only a critical question as the application of these policy instruments are not uniform even in the developed economies setting (there are some well-functioning Carbon taxing i.e., Canada, Japan some European countries, and Carbon trading schemes ie. in EU), but some can argue that adding a penalty to the under-financed state-owned energy companies would simply amplify the barriers in the electrification process.	We thank the reviewer for these comments. We are pleased they were able to gain a picture of how the technology mix changes as different policy interventions are tested. We strongly agree with the reviewer that reference to real-world policies was underdeveloped in the manuscript and have made changes outlined below. Firstly, on the point relating to the non-uniformity of policy interventions, we recognise this as an important point. In recognition of this, in our analysis, we conduct regional sensitivity to understand the level of financial support that there would need to be for impact. We have explicitly linked the point about non-uniformity directly when outlining the research questions in the final paragraph of the introduction, on Page 4: “And, finally, acknowledging the unevenness of existing policies and country heterogeneity, what is the sensitivity to different levels of policy intervention and how does this vary spatially?” We agree with the reviewer that adding a penalty for unmet demand could amplify barriers to electricity access. We now directly acknowledge this point in the discussion section on Page 13: “In practice, implementing a financial penalty for poor reliability, as is done in some high-income countries such as Norway, the Netherlands and the UK^{41, 42}, may not be feasible or desirable in countries where electricity supply companies are in a state of financial stress.” Partly in response to this point, we also outline how a performance subsidy for high-reliability systems could be applied instead, see further information below.

#	Comment	Response
	If the authors could provide some examples in some SSA countries (or from other developing countries) where these instruments were tried or introduced, it could increase the research added value to the policy agenda and decision making as well as to the ongoing research discussion. Otherwise, the well prepared analysis has the risk of remaining only an academic exercise.	Linking to policy examples is important and this was not well developed in the previous draft. In the discussion section on Pages 13-4, we have discussed policy examples for both carbon pricing schemes, and regarding reliability interventions. For reliability, the precedent of applying a penalty for outages in electricity supply comes from high-income countries. We have therefore detailed examples where penalties are charged to energy supply companies for power outages. Given the point raised above regarding the possible negative implications of applying a penalty scheme to cash-strapped distribution companies, we give an example from sub-Saharan Africa of a subsidy scheme paid to mini-grid operators for high-reliability systems: “In practice, implementing a financial penalty for poor reliability, as is done in some high-income countries such as Norway, the Netherlands, and the UK [41, 42] may not be feasible or desirable in countries where electricity supply companies are in a state of financial stress. Instead, offering an incentive scheme such as a performance-based subsidy for supply companies offering households a higher level of reliability may be a viable alternative. This has been demonstrated in Nigeria where some mini-grid developers are paid a subsidy per connection provided a minimum level of reliability is met [12, 43]. Policymakers should seek to design mechanisms that can ensure reliable electricity access options are implemented and thus reduce the negative impacts felt by households from unreliable electricity grids in many SSA countries.” We have included additional information in the Supplementary Materials demonstrating a way in which a reliability subsidy could be implemented that would have the same impact as applying a penalty. See section S2.4 and Figure S2 in the Supplementary Materials (Pages 27-8) On carbon pricing, we have included examples of carbon pricing schemes, both taxation schemes and carbon credit schemes in the above-referenced discussion section. There is a carbon tax scheme operating in South Africa, with several other countries in SSA considering implementing one. In addition, we give details (Pages 13-14) of two active initiatives that offer carbon credits to low-carbon electricity access projects. This may be an alternative way in which carbon pricing impacts the shares of technologies used: “As carbon markets become more prevalent globally, they are increasingly likely to have a positive influence on the cost competitiveness of low-carbon electricity access solutions such as PV-based mini-grids. With the exception of South Africa [44], countries in SSA presently do not have formal carbon taxation systems. However, other governments in SSA, such as Côte d’Ivoire, Botswana and Senegal are considering implementing schemes⁴⁵. An additional consideration is the voluntary carbon market. There are already examples where low-carbon off-grid electricity access projects are being developed with the help of financing from carbon credits [46]. Schemes such as the Carbon Initiative for Development (Ci-Dev) and the D-REC initiative are facilitating carbon finance support for low-carbon electricity access projects [33, 47]. Whilst not directly translatable from this analysis due to the complexities in carbon

#	Comment	Response
		crediting schemes, our analysis points to the impact carbon credits may have, depending on the price level and where they are implemented. Our analysis highlights where schemes may be most effective for supporting lower-carbon electricity access." Overall, with the inclusion of the above changes, we feel that the manuscript has a more policy-relevant discussion, and the analysis is now framed in a way that makes it more relevant to real-world considerations around different policy mechanisms.
3	Reviewer 2 The manuscript titled 'How reliability and carbon prices impact pathways to universal electricity access in Africa' addresses an important and relevant topic. However, there are several areas where the manuscript needs improvement to enhance its clarity and overall quality.	We thank the reviewer for their comment and are glad they believe that it is an important and relevant topic. We have, where possible, addressed the reviewer's comments (see below), and we hope that they agree that it now has enhanced clarity and quality.
4	The title of the manuscript, "How reliability and carbon prices impact pathways to universal electricity access in Africa," does indeed sound like a question. It would be more appropriate to rephrase it into a statement or summary of the study's objective. For example, "Assessing the Impact of Reliability and Carbon Prices on Achieving Universal Electricity Access in Africa."	We thank the reviewer for their useful comment. We agree the previous title was unclear in its wording. We have changed the title to: "Reaching Universal Electricity Access in Africa: A Geospatial Analysis of System Reliability and Carbon Price Impacts"
5	One of the primary issues with this manuscript is its lack of coherence and proper organization. The structure of the paper needs to be improved for better readability and comprehension.	We thank the reviewer for their comment and hope that we have improved the structure and readability. We have added to the Introduction section, added further literature, improved clarity in the methodology and the results, added to the discussion section and fixed errors in the references. Please see more detail on each of these points below. Some aspects of the structure are due to the formatting requirements of the journal and cannot be adjusted, these are highlighted below.
6	Introduction: Provide a clear introduction that outlines the importance of the study, its objectives, and the research question.	We have made changes to the Introduction section to emphasise the importance of the study and to detail the objectives and the research questions more clearly. Please see the last paragraph of the introduction for this section, on Pages 3-4: "This paper addresses the following four main research questions regarding least-cost pathways to universal household electricity access in SSA. Firstly, how might the shares of each technology change at the aggregate and disaggregate (country) level as the electricity demand changes? Secondly, how does implementing a policy that includes a cost for units of unmet electricity demand change the share and spatial distribution of each technology used? Thirdly, how might implementing a carbon price change the shares and spatial

#	Comment	Response
		distribution of the technologies used? And, finally, acknowledging the unevenness of existing policies and country heterogeneity, what is the sensitivity to different levels of policy intervention and how does this vary spatially? In addressing these questions, this paper aims to catalyse further research in this area and give a basis for implementing policies that can both expand access via low-carbon technologies and improve the value of access given to households from improved reliability of systems.”
7	Literature Review: Expand the literature review section to provide a more comprehensive overview of existing research in this area. This will help readers understand the context of the study.	We thank the reviewer for this suggestion. We have added four further references to the literature review (Introduction section) and we hope this gives a broader picture of existing related research, and the context of the study. The Nature Communications formatting guidelines suggest no more than 70 references, so we are limited in adding further detail given that we have also added references to the discussion section. Please see the Introduction Section on Pages 1-3.
8	Methodology: The methods section should be expanded to provide sufficient detail for the work to be reproduced. Explain the data sources, variables, and statistical methods used in the analysis.	We thank the reviewer for this comment. We have gone through the Methods section and enhanced clarity regarding some of the sections that we deemed potentially less clear. See, for example, improved clarity to the section with the subheading “Estimation of baseline population without access”. Additionally, three subheadings within the methods have been updated to better reflect the contents and make things clearer for the reader. Regarding sources of data, within the Methods section, we have a section “Geospatial Data” that details the geospatial datasets used for the analysis. Some of the detailed modelling data inputs, and the detailed model description, are included in the Supplementary Materials due to the formatting restrictions (Methods word limit). Please see the updated Methods section, Pages 14-20, and the Supplementary Materials, which starts on Page 26. We now feel that all major sources of data, tools used, and methodological choices are sufficiently clear, having compared to other papers in the Nature Communications journal.
9	Results: Present the results in a logical sequence and use clear headings and subheadings to guide the reader through the findings. Visual aids such as tables and figures should be used to enhance understanding.	We thank the reviewer for their suggestion here. To improve clarity, we have updated all the results subheadings to improve readability and flow. They now read as follows to reflect the different things presented: “Progress towards universal household electricity access and the baseline scenario” “Electricity demand level substantially impacts technology choices for electricity access” “Inclusion of policy interventions accounting for variation in electricity system reliability” “Inclusion of a carbon price scheme”

#	Comment	Response
		“Additional investment and emissions for universal household electricity access by 2030” In addition, we have added further detail to several of the figure descriptions to ensure that the visual aids are clearer to the reader. See longer and clearer figure descriptions for Figures 2, 3, and 4, seen on Pages 7, 8, and 11
10	Discussion: Analyze the results in the context of the research objectives. Discuss the implications of the findings and how they relate to existing literature.	We thank the reviewer for this comment. We have extended the Discussion, adding additional text to all three main sections. We now go into more depth regarding the real-world relevance of the results, focusing especially on the carbon pricing and reliability aspects of the paper, which are the most novel areas it covers. Please see the updated Discussion section on Pages 12-14. In addition, please refer to the response to comment 2 in this document which provides some further detail regarding the changes to the discussion.
11	Conclusion: The manuscript lacks a conclusion section, which is essential to summarize the key findings and their significance. Provide recommendations for future research or policy implications.	Nature Communications journal formatting guidelines do not permit a conclusion section. Included in the Discussion section we focus on what we feel are the most salient research findings and have included a discussion of both future research suggestions and policy implications. We hope the reviewer feels satisfied that the discussion section covers the points they would hope to see there.
12	References: Ensure that all references are cited correctly and consistently throughout the manuscript.	We thank the reviewer for pointing out these issues and have gone through the references and reference list to ensure everything is properly presented and consistent.
13	In conclusion, while the topic of the manuscript is noteworthy and relevant, it requires significant revisions to improve its overall quality. The authors should focus on enhancing the structure and coherence of the paper, providing sufficient detail in the methods section, and rephrasing the title to make it more declarative. These revisions will enhance the manuscript's readability and impact. I recommend that the authors carefully address the above-mentioned issues and consider incorporating the suggestions into their revised manuscript.	We thank the reviewer for their comments, and we agree that these revisions have aided clarity and will make the paper easier to read.

REVIEWERS' COMMENTS

Reviewer #1 (Remarks to the Author):

The changes the authors introduced answer my previous concerns on applicability of carbon pricing in SSA setting to the extent possible. The authors agree that there are only few initial trials in developing setting.

In this topic the authors used already 2 relevant IEA documents.

I would recommend to the authors to include and refer the recent very relevant IEA report on "Financing Clean Energy in Africa" 2023. IEA calculates that reaching a Sustainable Africa Scenario would require more than doubling the present level of energy investments. They express the importance of support schemes. It claims that "only around half of the new electricity access connections providing the most basic energy services are likely to be affordable in the absence of additional financial support, such as subsidies, grants or tariff reform."

The authors could argue that in this setting carbon pricing could play an important role in the investment decisions. In fact in the chapter on "Mobilising capital for a sustainable future" the IEA talks about the importance of carbon markets.

"Carbon markets can attract investment by supporting project revenue streams for a variety of projects. ... However, to ensure the effectiveness of carbon markets, countries first need to adopt solid regulatory, monitoring and verification frameworks."

Response to Reviewers: NCOMMS-23-28925-T (final revisions)

#	Comment	Response
1	Reviewer 1 The changes the authors introduced answer my previous concerns on applicability of carbon pricing in SSA setting to the extent possible. The authors agree that there are only few initial trial in developing setting.	We thank the reviewer for their feedback and are glad that our changes address previous concerns regarding the applicability and existence of carbon pricing in SSA.
2	In this topic the authors used already 2 relevant IEA documents. I would recommend to the authors to include and refer the recent very relevant IEA report on "Financing Clean Energy in Africa" 2023. IEA calculates that reaching a Sustainable Africa Scenario would require more than doubling the present level of energy investments. They express the importance of support schemes. It claims that "only around half of the new electricity access connections providing the most basic energy services are likely to be affordable in the absence of additional financial support, such as subsidies, grants or tariff reform."	We thank the reviewer for this reference as we were not aware of it. It strengthens the paper and we have added it in the introduction on page 3 along with the following text: "Given that a significant share of new connections required to meet universal access are unlikely to be financially viable without additional financial support, carbon financing may play an important role in expanding access [36]" And in the discussion section, this is outlined below.
3	The authors could argue that in this setting carbon pricing could play an important role in the investment decisions. In fact, in the chapter on "Mobilising capital for a sustainable future" the IEA talks about the importance on carbon markets. "Carbon markets can attract investment by supporting project revenue streams for a variety of projects. ... However, to ensure the effectiveness of carbon markets, countries first need to adopt solid regulatory, monitoring and verification frameworks."	We have also added reference to this IEA document in the discussion section on page 14, highlighting the fact that the right frameworks are needed to see the full potential of carbon markets come to fruition: "To maintain effective carbon markets and realise their possible benefits for energy access, countries in sub-Saharan Africa need to work to build the regulatory, monitoring and verification systems. If these can be implemented, our analysis highlights where schemes might be most effective for supporting lower-carbon electricity access."